# Experiences and Development Impacts of Securing Land Rights at Scale in Developing Countries: Case Studies of China and Vietnam

**Frank F. K. Byamugisha**

Independent Consultant, Washington, DC 20433, USA; fkbyamugisha@gmail.com

**Abstract:** This paper reviews experiences and development impacts of a selected number of developing countries in Asia and Africa that have used emerging land registration approaches to rapidly secure land rights at scale. Rapid and scalable registration is essential to eliminate a major backlog of the world's unregistered land, which stands at about 70 percent. The objective of the review, based on secondary data, is to draw lessons that can help accelerate land registration across many countries. While the focus is on China and Vietnam, the findings are buttressed by those from previous reviews in Ethiopia and Rwanda. The registration approaches used in these four countries were found to be cost-reducing, fast, inclusive and scalable enough to secure land rights for all within one generation. They also had significant positive impacts on land tenure security and investment. In addition, they indirectly along with other economic reforms contributed to rapid economic growth and a reduction in extreme poverty. The experience from these Asian and African countries offers important lessons including the need for strong political commitment and to develop flexible legal and spatial frameworks that fit the purpose of land registration, instead of the rigid technical standards set by land professionals.

**Keywords:** securing land rights; land registration; development impacts; fit-for-purpose land administration

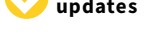



## 1. Introduction

The role of property rights in the economic development of the Western world has been well documented by economic historians and development economists including North and Thomas [1] and Rosenberg and Birdszell [2]. Well-defined and enforced property rights are key to economic development. Property rights in land are defined and enforced within formal governance structures called land administration systems. In Western European countries, virtually all their land, more than 95 percent, is registered in land administration systems, as reported by Schmid and Hertel [3]. However, the vast majority of the world's land, about 70 percent, remains unregistered and administered outside formal land administration systems [4]. To accelerate economic development and eradicate extreme poverty, especially in developing countries, it is important to register land quickly, at scale and in sustainable land administration systems.

In many developing countries, systems of land registration were initiated more than a century ago based on Western-style approaches that used rigid, high accuracy and skill-intensive standards of land surveying that are too costly to scale-up and maintain. Consequently, national registration coverage of land parcels (and owners) has been low, and limited primarily to urban areas and selected high-value rural areas. Moreover, subsequent transactions have not always been registered due to high registration charges relative to expected benefits, thus rendering the land registration records outdated. These conventional land registration approaches and their associated high costs have been at the root of the low levels of formal documentation and administration of land rights in many

developing countries, including those in sub-Saharan Africa, where only about 10 percent of rural land is registered [5].

With new cost-reducing technologies emerging, an increasing number of countries has documented their land rights faster and at scale, including former Soviet Union countries after 1990 [6] and Thailand after 1984 with its 20-year land titling program [7]. A handful of countries have followed suit in Asia, Latin America and Africa, but they have struggled to document the more challenging lands in urban informal settlements, state lands, pastoral rangelands, woodlands and forestlands.

To register land at scale, including the challenging areas indicated above, new approaches have been developed including the fit-for-purpose approach to land administration. The approach has been embraced by key development players such as the International Federation of Surveyors (FIG) and World Bank [8] as well as the United Nations Food and Agriculture Organization (UN FAO)-led *Voluntary Guidelines on the Responsible Governance of Tenure of Land, Fisheries and Forests in the Context of National Food Security (VG-GTs)*. Global Land Tool Network (GLTN)/United Nations Human Settlement Programme (UN HABITAT) supported the development of guidelines for its implementation at country level [9]. The emerging cost-reducing, affordable, fast and scalable approaches hold considerable promise to accelerate registration and improve land administration. However, a thorough evaluation of implementation experience is required to draw lessons that can inform adoption across many countries, building on some previous evaluations [5–8,10–12]. This article is intended to supplement these evaluations.

In addition to technical evaluations, socio-economic evaluations of land registration initiatives have also been done to assess their development impacts. A review of this literature suggests a positive impact of documentation of individual land rights on investment and productivity although gains have on average been more modest in Africa but stronger in Latin America and Asia [13,14]. The weaker gains in Africa have been attributed to the pre-existing context there, primarily the predominant customary land tenure having been relatively secure before formalization (hence weaker productivity gains from formalization); and the operating environment there (with less developed financial markets, and weak infrastructure and complementary investments) being inadequate to support a robust response of investment and productivity. Overall, these socio-economic evaluations suggest that documentation of land rights has significant impacts on investment and productivity, but context and complementary factors matter.

This paper reviews experiences of two Asian countries, China and Vietnam, in securing land rights at scale. The two populous nations have many rural arable land parcels, 1.5 billion in China and 70 million in Vietnam (see Table 1). They initiated country-wide documentation of land rights in the 1980s and 1990s using principles similar to those of fit-for-purpose approach to land administration long before it was articulated and formalized. Yet, there has not been a comprehensive and structured review of how the land registration was done and its results and development impacts. The objective of this paper is to review their land registration experiences, draw lessons and identify remaining challenges. The findings from the review are cross-checked against those from previous reviews of experience in two African countries, Ethiopia and Rwanda, to strengthen the conclusions and to broaden the global relevance of the lessons learnt.

**Table 1.** Population and rural land parcels of case study countries.

| Country | Population Est. 2020 (Millions) | No. of Rural Arable Land Parcels | |
|---------|---------|---------|---------|
| | | Year | Millions |
| China | 1439 | 2008 | 1500 |
| Vietnam | 97 | 2002 | 70 |
| Ethiopia | 115 | 2019 | 50 |
| Rwanda | 13 | 2012 | 12 |
| Total | 1664 | | 1632 |

Source: Estimates of population are based on data from the World Bank Database [15]. Rural land parcels data for China, Vietnam, Ethiopia and Rwanda were obtained from [12,16–18].

## 2. Approach and Methodology

This is a desk review that addresses the following research questions. How was land registration done in case study countries and what was the national coverage by land types and rights? How do the underlying principles to land registration relate to those in the fit-for-purpose land administration? What were the development impacts notably on investment, productivity, economic growth, poverty reduction and gender equity? To address these questions, the review covers four areas: (i) the legal framework to register land rights; (ii) the documentation, registration and certification of land rights; (iii) the development of unified and sustainable registration systems; and (iv) the development impacts of securing land rights at scale. The first two areas are important in assessing whether the legal framework was flexible enough and the registration approaches participatory, affordable, fast and scalable to secure land rights for all. The results would determine if the guiding principles of fit-for-purpose land administration are applicable retrospectively to the approaches that were used in those countries. The third helps in assessing whether the registration systems are unified or adequately coordinated to avoid high costs of establishment and maintenance, and to preserve quality, integrity and consistency of land registration data. The fourth area is important to justify investments in securing land rights.

Research for this paper was based on secondary data obtained through internet search and also from the records and databases of the World Bank where the author used to work and currently works as a consultant. Additional data was obtained from (a) government and donor-funded program and project documents including design, appraisal and evaluation documents, (b) publications by researchers and development practitioners, and (c) publications and databases of global development agencies and other actors. The method of research is qualitative based on case studies. The assessment of land registration and tenure security impact on economic growth is done by relating the former to the latter through intermediate outcomes such as investment and agricultural productivity since there is no direct relationship between land tenure security and economic growth. Past attempts to estimate their relationship have faced major problems of attribution [19]. Hence, we have assessed the impact of land registration on economic growth based on previous empirical research findings on registration impacts on investment and productivity. This has been supplemented by an assessment of economic growth (and poverty reduction) trends over the period of land registration interventions. Assessment of gender equity was based on reported land registration data disaggregated by gender.

## 3. Assessment and Results of Case Studies

### 3.1. China

Since the establishment of the People's Republic of China in 1949, the country relied on a centrally planned socialist economic system for its development until 1978 when it began a program of gradual but fundamental reform of the economic system toward a market-based mixed economy that continues to the present day. But even under the reform

program to date, land has remained under state ownership in urban areas and under collective ownership in rural areas where farmland is contracted to households [20]. Hence, the registration of contractual rights to land plays an important role in confirming the land rights of users. With land tenure being central to the reform of the socialist economic system, there have been a strong political commitment to land tenure reform at all levels of government.

Land issues being local by nature, the decentralized system of government from central to provincial, municipality, prefecture-city and county down to commune level has enhanced participatory engagement especially in the registration and certification of land rights [21]. In addition, the securing of land rights has benefited from the fundamental reform of the socialist economic system in that the legal framework has had to be developed afresh and flexibly to cater for a multiplicity of reforms geared to achieving a market-based mixed economy. This provided opportunities to try out pragmatic approaches to land registration with principles similar to those of fit-for-purpose land administration [9].

### 3.1.1. The Legal Framework to Register Land Rights

From 1978, when it initiated major reforms to dismantle the collective system of production in favor of production by households through a Household Responsibility System (HRS), China's land tenure reform has followed a steady path to develop a comprehensive and flexible legal framework for land administration [20].

**Rural Land Rights**

Under the HRS and with rural land still collectively owned, rural households accessed land for farming through contract from local commune authorities, initially for five years, then extended in 1984 to 15 years, and to 30 years in 1993 when transfers of contracts for value among households within the collective were permitted but with prior consent from the collective. In addition, restrictions were imposed on periodic readjustment of contracted farmland which had been necessary to accommodate population growth within communes [20]. In the revisions to the Land Administration Law in 1998 [22], farming households were granted long-term and guaranteed land use rights which, in 2013, were confirmed to stay unchanged for a long time. The law also required farmers to receive a 30-year land use contract certificate [23]. In 2002, the land rights over farmland were strengthened by permitting households to transact them freely within their collective and also to sub-lease to households outside the collective. Contractual terms for grassland were set at 30–50 years, and for forestland at 30–70 years while they were undefined for residential rural land but with buildings on it owned in perpetuity. However, residential land could not be mortgaged or transferred to urban residents, to households outside the collective or used for non-agricultural development [24].

In 2007, the land use rights were further strengthened by clarifying that they were no longer contractual rights but property rights whose 30-year term were extendable upon expiration; it also laid out and strengthened arrangements for property registration [25]. The strengthened provisions for registration of farmland were reiterated and expanded in 2010 to also cover forestland and residential land, and to establish a rural land registration system. In 2013, the central authorities set a target of completing registration of rural land rights within five years. In the Decision of the Communist Party in November 2013 and in the newly-revised Rural Land Contract Law in 2019, the mortgaging and guaranteeing of contracted farmland as collateral were confirmed [24].

Rural land, collectively owned, is contracted out for farming (farmland) while grassland and forestland are contracted out for other rural development enterprises. The remaining land is construction land which is under two categories; land on which residential houses are built for the households (residential or homestead land) and also used to meet demand for village towns and enterprises; and for urban expansion. Construction land for residential houses, village towns and enterprises is allocated by collective authorities while land for urban expansion is expropriated by local governments with payment for

compensation, and added to the stock of urban construction land for urban expansion or contracted out for private sector development [21].

**Urban Land Rights and Buildings**

Urban land is state-owned and managed by local governments. Land for construction is used by local governments for urban development, including infrastructure and other public services, while the remaining land is contracted to institutions and individuals for residential, industrial and commercial development. The contracting of land, through land market auctions, is for 70 years for residential purposes; 50 years for industrial development projects; and 40 years for commercial, tourism, or recreational purposes. The contractual terms are renewable, transferable and mortgageable and, for individuals, inheritable [26].

Prior to 1988, there were no individual rights to urban land in China. The state had a monopoly on housing, and distributed houses to households through their work units and enabled them to obtain accommodation at low rents. Since there was only public landownership at that time, there was little need to introduce a full-fledged land registration system. In 1988, China introduced a policy of privatizing its housing and of creating a housing market. It changed its constitution to allow individual urban land use rights and building ownership, and their transferability. While rules, laws and guidelines for registering urban private housing were initiated as early as 1982, they were only gradually developed and consolidated in the Real Estate Administration Law of 1994 which premised legal ownership of private housing on registration and award of a certificate for building ownership and another award of a land use rights certificate for land on which the property is built [27]. Subsequent laws and regulations were promulgated to fine tune the details and processes of registration of real estate transactions including mortgaging, starting with the Property Law of 2007. The latter, in particular, strengthened property rights by tackling deficiencies in the immovable property registration system by providing a national vision and framework to guide local registration offices [26].

3.1.2. Documenting, Registering and Certifying Land Rights

**Rural Land Rights**

There are two sets of documentation required to register and certify rural land rights. The first is textual or alphanumeric data that records rights in land. The second is a spatial framework in the form of cadastral or map data that show boundaries and extent of land over which these rights apply. Until the adoption of the 2007 Property Law which required and provided for the registration and certification of property rights, the registration of rural land rights (contractual farmland, grassland and forest land; residential or homestead land; and rural construction land) was based mostly on textual data, without a requirement for a spatial framework. Registration was required by the 1998 Land Administration Law to confirm the 30-year contractual term with certificates, referred to as standardized and notarized contracts [23]. The applications for certification of contractual rights for various kinds of rural land were submitted to the responsible agencies for processing and registration at county level as required by their statutes: to Ministry of Agriculture for registration of farmland and grassland; and to the State Bureau of Forestry for registering forestland. The Ministry of Land and Resources (MLR) was responsible to register other rural collective lands and to oversee the registration of other lands [28]. The registration and certification was based mostly on textual ownership data, without using a precise spatial framework [21,29].

After adopting the 2007 Property Law and based on its provision for national standards for registration, MLR in 2007 issued detailed measures to provide not only textual data on contractual rights but also spatial data that indicate the geo-position, boundaries and size of land covered by the rights as well as the methods of recording and indexing, and the powers of review by the registrar [21].

In 2009, the central government initiated measures to undertake pilot projects in the country, building on a land registration experiment in Anhui province started in 2005 [23]. Each province was ordered by the central government to continue promoting these land

registration projects. The provincial governments were given the responsibility to organize and lead, the city-level governments to organize and coordinate, and the county and township governments to implement these projects ensuring that they are completed in every village in their jurisdictions [21].

The projects piloted an area by area systematic and participatory approach using commune leaders and members (with little training) to demarcate, adjudicate and register land use rights. They used textual data of contracted rural land rights and spatial data mostly from high-resolution satellite imagery (0.5 m resolution), supplemented by more precise ground survey data in case of high value land, to document and register contractual land use rights. They also introduced digital land records [30,31]. The piloting continued through 2012 and, in 2013, it was scaled up to cover the whole country. The procedures used in the scale-up included: publicity and mobilization campaigns, and short-training of local non-professionals in land demarcation and adjudication by surveyors and other land professionals; survey of households and confirmation of their membership in and contractual rights obtained from the collective; ortho-rectified aerial imagery (0.4 or 0.5 m resolution) supplemented where necessary with detailed survey of boundaries using a variety of survey instruments; public display and verification of field survey and adjudication results; registration of land rights; and issuing of certificates of land use rights [21,31]. While the piloting and scaling up of registration was the responsibility of county and township governments for all villages in their jurisdictions, it's the land professionals in MLR in the counties and townships that were responsible for training and supervision of non-professionals (mainly commune leaders and members) to do the demarcation and adjudication, while also undertaking quality control [21,28].

The scaling up of registration in 2013 was done rapidly while also expanding the capacity and national coverage of local Land Administration Offices under MLR (restructured and renamed Ministry of Natural Resources in 2018) to support first time registration and to also handle subsequent land transactions in a computerized environment. By the end of 2018, China had virtually completed first-time registration and certification of rural contracted land, which was 1.48 billion mu (one mu is equivalent to 0.165 acres) or about 98.87 million hectares, accounting for 89 percent of the measured area of contracted land [21].

From 2015, the scaling up of land registration was done hand-in-hand with the implementation of a nationwide, unified real property registration system which was completed in mid-2018. It included organizational consolidation of registration responsibilities under MLR and the development of a national digital information platform to share land information and enhance collaboration across all stakeholders [32].

**Urban Land Rights and Buildings**

While the legal framework to support registration of urban land rights was initiated in 1982, actual registration commenced in 1988 when an economic policy of privatizing housing and a housing market were introduced, and constitutional changes made to allow transferability of privatized land use rights. The main requirement for registration was textual documentation of land use rights. An additional requirement for a spatial framework was introduced later with the adoption of the 2007 Property Law which also provided for national standards to register urban land rights (urban construction land; and contractual land use rights and the associated developments). The Ministry of Housing and Urban-Rural Development (formed in 2008; replacing Ministry of Construction) followed up in 2008 with the issuing of detailed measures of registration including requirements for textual data on land use rights and spatial data. The required spatial data were geo-position, boundaries and area of the immovable property captured in updated cadastral maps based on orthophotos (0.2 to 0.4 m resolution), high resolution satellite imagery (0.5 m resolution), drones and detailed ground surveys [26].

Unlike the documentation of rural land rights which followed an area by area systematic registration approach, the registration of urban land rights was done sporadically based on applications from rights holders. It should be noted that the registration of

building ownership was done separately from land use rights, with building ownership registered at the local Authority for Housing while land use rights and related transactions such as mortgages were registered at the local Land Administration Authority, resulting in the issuing of separate building ownership and land use rights certificates, respectively. With commencement of registration of contractual urban land rights in 1988, pressure was put on both the local Authority for Housing and the local Land Administration Offices to expand their capacity and urban coverage to handle the registration demand which increased with the boom in real estate that ensued in the 1990s and 2000s following housing privatization and development of the housing market [33]. The registration of urban land and properties was based on cadastral maps with scales of 1:500 and 1:1000 in the larger cities and 1:2000 in the smaller cities and other developed areas [26,34]. They also initiated the digitization of land records and to operate in a computerized work environment [35]. The registration of land and buildings, combined with the incorporation of peripheral and urban villages (informal settlements) in urban master plans [26], reduced the percentage of the urban population living in slums from about 44 in 1990 to about 25 in 2018 [36].

### 3.1.3. Developing a Unified and Sustainable Registration System

The Property Law of 2007 provided a vision of a unified land and property registration system to overcome fragmentation in registration of property rights which was leading to inefficiencies and increased incidences of error. Fragmentation had three sources: property registration being done at three separate levels of government; building ownership being registered separately from land use rights; and different rural land types (farmland, grassland and forestland) being registered in separate systems and agencies. But the law lacked regulations and implementation measures to achieve the vision, and hence was not implemented until 2015 when they were put in place [33]. But some jurisdictions could not wait that long and so some municipalities, provinces and prefecture-cities came up with their own local regulations either modifying or supplementing the national regulations while others made much more profound change to the national framework [33]. For example, Shanghai Municipality in 2009 overhauled its Regulations on Registration of Real Estate to implement a unified registration of land and buildings thus issuing unified certificates of right to land and buildings [35]. A few other municipalities followed suit including Chongqing and Tianjin, in 2004 and 2006, respectively. On the other hand, Beijing municipality, decided to continue with registration of building ownership while discouraging registration of land use rights except for an exclusive list of residential developments. Some prefecture cities also did likewise including Dalian of Liaoning Province, Qingdao of Shandong Province, Xiamen of Fujian Province and Kaifeng of Henan Province [33].

In November 2013, MLR was assigned the responsibility to guide and supervise the implementation of the unified property registration system throughout China. Accordingly, MLR established in August 2014 a new department, the Bureau of Real Estate Registration and put it in charge of implementing the unified property registration system [37]. In November 2014, it issued Interim Regulations on Real Property Registration and commenced its implementation in March 2015 with a plan to establish a nationwide, unified real property registration system in three years [33]. Through 2015 and 2016, Real Estate Registration Bureaus were established in various provinces to implement the unified registration system [38]. Based on the experience gained by some municipalities and prefecture-cities that had implemented unified registration systems on their own, a national digital land information platform was developed to support the integration of real property registration data of buildings, urban land, farmland, woodland, grassland and forestland of varying quality, under one system [35]. According to the Ministry of Natural Resources, the unified real property registration system came into effect on 18 June 2018. It started connecting "3001 property registration stations in 335 cities and 2853 counties serving more than 300,000 enterprises and individuals on average each day, according to latest statistics" [32]. With the national unified real property registration system in place and operational, the quality and efficiency of registration services are expected to improve

throughout the country as they did in Shanghai and other jurisdictions which implemented unified property registration initiatives on their own before the introduction of the national program in 2015.

### 3.1.4. Development Impacts

Implementation of the HRS had a significant positive impact on investment and agricultural growth as reported by Lin [39]. Also, according to impact studies, the accompanying first round certification of contractual rights to farmland had a significant positive impact on investment and productivity for the farmers whose land was not subjected to frequent reallocation by commune authorities; the impact was achieved through increased investment incentives, renting-out land and migrating out of farming to more rewarding economic activities [23,40,41]. The second-round certification, both piloting and scaling up, also significantly promoted investment incentives among farmers who had not had land reallocation experience but negatively affected those who had experienced big reallocations [42].

Notwithstanding the dampening effects of land reallocations on investment response to certification of contractual land rights, the legal clarification and registration of land rights together with other economic reforms and public investments introduced after 1978 had a profound impact on China's economy as reported by some researchers [43]. Over the 40 years of economic reforms (1980–2019), China's economy achieved an average annual real growth rate of 9.4 percent measured in GDP constant prices, based on calculations using data from the IMF World Economic Outlook Database [44]. Land registration and tenure security contributed to the growth indirectly mainly by stimulating private investment while the greater contribution to growth came from the other major economic reforms and public investments [43]. The rapid and sustained economic growth contributed to reducing the proportion of people living below the poverty line (US$1.25 per day) from over 85 percent in early 1980s to about 0.5 percent (using the poverty line of US$1.90 per day) in 2016, according to data from the World Bank Poverty and Equity Data Portal [45].

### 3.2. Vietnam

After its 1954 independence from the French, which left it divided into two parts (North and South), Vietnam emerged from the "Vietnam war" in 1975 with a reunified country. In 1986, it introduced sweeping economic reforms (the "Doi Moi" policy) and a move from a socialist economic system towards a market-oriented economy, including the dismantling of collective production and the allocation of land use rights to households [46]. Like in China, where land tenure was central to the fundamental reform agenda, there was a strong political commitment to land tenure reform at all levels of government in Vietnam. And reform implementation, including land registration, was participatory, aided by "peoples committees" which operate at every decentralized level of government from provincial to district and down to the commune level [46]. In addition, the legal framework was developed afresh and flexibly to cater for a multiplicity of sweeping economic reforms. This created opportunities to try out pragmatic approaches to land registration that are underpinned by principles similar to those of the recently formalized fit-for-purpose land administration [9].

### 3.2.1. The Legal Framework to Register Land Rights

Since 1986 and the "Doi Moi" policy, Vietnam progressively moved towards a market economy, with two major changes in rural land rights. In 1988, the collective system of production was dismantled in favor of production by households. In the agricultural sector, Resolution 10 of the 1988 Land Law granted land use rights (for 10 to 15 years) to individual households, with land allocation done by commune authorities. Land use rights could not be traded or exchanged until adoption of the 1993 Land Law which assigned five rights to land users-to transfer, exchange, lease, inherit and mortgage land use rights.

Ownership was vested in the entire people, according to the 1992 Constitution, with the state managing the land on their behalf [47].

The duration of rural land use rights was increased to 20 years for annual crops (increased further to 30 years in a 1998 revision) and to 50 years for perennial crops and forestland. The law also enabled authorities to allocate land for urban residential use on a stable long-term basis while land used for income production could be allotted or leased for short periods based on business production plans. It also provided for registration of land use rights.

These land rights were extended and clarified by the 1998 amendment to the Land Law to allow sub-leasing and Vietnamese entrepreneurs to contribute land rights to joint ventures with foreign companies. Further amendments in 2001 simplified procedures in urban areas to allow foreign investors to lease land for renewable periods of 50 to 70 years [47].

The 2003 Land Law provided a legal boost to the emergence of a market for land use rights. It also provided further equality on land rights between domestic and foreign investors, including Vietnamese permanent residents abroad, to buy property associated with their land use rights, and between husband and wife in certification of land use rights [48].

The 2013 Land Law extended lease terms for all agricultural land to 50 years for both annual and perennial crops, and broadened the scope and duration of land rights by allowing landholders to transfer land as gifts to others or as shares in joint ventures [49].

The land rights for rural residential and urban land were also reformed. For residential land (for houses) in rural and urban areas, the allocation to households and individuals is for "long term use", basically indefinite duration. The 1992 constitution and the Civil Code give citizens the right of housing ownership in both rural and urban areas [50]. The Housing Law and Real Estate Business Law of 2014 extended "land-use rights" to foreign investors, allowing title holders to conduct property transactions, including mortgages [51].

### 3.2.2. Documenting, Registering and Certifying Land Rights

Following introduction of the "Doi Moi" policy, Vietnam started registering land rights and issuing land use rights certificates (LURCs) to households in 1994 on the basis of the 1993 Land Law [46]. This was done based on textual data on land rights, mainly the land allocation approval package of documents including the application for LURCs by individual households, evidence of support by neighbors and approvals by the commune and district peoples committees and by the district land allocation and registration committees. Cadastral maps of varying quality (some with inadequate scale; others outdated) [52] indicating location, boundaries and size of land, were also used in some cases [47,53].

The registration process of land rights involves all levels of government, from central down to commune level, and non-professionals (with short training) especially members of commune and district peoples committees. At central government level, the General Department of Land Administration (GDLA), established in 1994, is responsible for registration and issuing LURCs for rural land. After the Ministry of Natural Resources and Environment (MONRE) was established in 2002, its departments absorbed the functions of GDLA. GDLA was re-established in 2009 and took over daily oversight of land administration under MONRE's guidance [46,54]. The Ministry of Construction had responsibility to issue building ownership certificates in urban areas. At regional and local level, the peoples committees are responsible for implementing land administration according to the laws, working with the Department of Natural Resources and Environment (DONRE) including land administration offices [46,54].

In 1994, the GDLA arranged with Ministry of Construction to sponsor the joint issuing of land use rights and building ownership certificates for urban land. This was formalized through amendments to the Land Law in 1998 to develop a unified system to register land use rights and ownership of the attached properties; the amendments laid out the associated processes and procedures for registration. At provincial and city levels, the

respective housing departments supervised by GDLA and Ministry of Construction were merged under the supervision of GDLA thereby enabling GDLA to take responsibility for issuing not only LURCs but also the Building Ownership and Land Use Certificates (BOLUCs) [53].

The registration of rural land was very rapid but slow for urban land. By the end of 2000 (after only six years of implementation), 90 percent of rural land users had been issued with LURCs while in urban areas, only 16 percent of land users had been issued LURCs [53]. Rural land registration was compulsory and undertaken systematically, area by area, while registration in urban areas was voluntary and done on request. Low registration in urban areas was attributed also to high land allotment fees (up to 20 per cent of the land values) accentuated by complex procedures of registration and incomplete cadastral mapping [50]. The issuing of LURCs in rural and urban areas was based largely on textual data, with hardly any precise cadastral data. The requirements were consistent with the urgency and national scale of the task to register an estimated 70 million land parcels in rural areas [17].

Following the adoption of the 2003 Land Law, Vietnam embarked on registering land rights using integrated textual and spatial data based mostly on orthophotos (0.4 m resolution) and high resolution satellite imagery (0.5 m resolution). In addition, an initiative was made to start issuing LURCs on a parcel (instead of household) basis to overcome challenges of registering subsequent transactions [55].

The cumulative total number of LURCs and BOLUCs issued after the adoption of the 1993 Land Law was about 30 million in 2006, according to reports by GDLA, indicating an annual average of 2.5 million LURCs/BOLUCs. As of December 2007, about 82 percent of agricultural land area, 62 percent of urban residential land area, 76 percent of rural residential land area and 62 percent of forest land area had been covered with LURCs [52]. The high volume of registration notwithstanding, the quality of registration documents was inadequate, with cadastral maps mostly inaccurate and outdated [52]. It was estimated by the World Bank that, to complete the nationwide registration including updating cadastral maps (at scales of 1:500 to 1:2000 in urban areas; and 1:2500 to 1:10,000 in rural areas), about 20 million of LURCs/BOLUCs would need to be issued or re-issued [52,56]. Much of this work was supported by a World Bank project, from 2008 to 2015, including the registration and issuing of 3 million LURCs, out of which 80 percent were certificates reissued with an upgraded spatial framework based mostly on orthophotos (0.4 m resolution) and high resolution satellite imagery (0.5 m resolution), supplemented by ground surveying [57]. Field adjudication and demarcation work was done mostly by local people, mainly members of communes and peoples committees, trained and supervised by qualified surveyors from GDLA supported by contract staff, while the back-office document processing was done entirely by GDLA land professionals supported by contract staff [57]. GDLA had about 12,000 land professionals (called cadastral staff) supporting the land registration program in the entire country as of 2015 [54]. Approximately 62 percent of the LURCs issued under the project were registered in the names of women or joint spouses [57].

Using support from development partners led by the World Bank, Vietnam was able to ramp up the quality and quantity of registration of land use rights from 2009 to 2015. As of January 2015, Vietnam had virtually completed its nationwide program of good quality land registration, with the following outputs and coverage of first time land registrations [54]:

- 20.2 million LURCs for agricultural land, representing 90% total area;
- 2.0 million LURCs of forestland, representing 98% total area;
- 13.0 million LURCs of rural residential land, representing 94% total area;
- 5.3 million LURCs of urban residential land, representing 97% total area; and
- 0.3 million LURCs for special purpose land, representing 85% total area.

According to MONRE, as of September 2018, the government had issued LURCs for 96.9 percent of land in Vietnam [51]. The registration of urban land and property, combined with the formalization of urban informal settlements, reduced the percentage of the urban population living in slums from about 61 in 1990 to about 14 in 2018 [36].

### 3.2.3. Developing a Unified and Sustainable Registration System

Following the 2003 Land Law that required a unified registration system in terms of organizational arrangements and operations, Vietnam established unified land registration offices at the provincial level in all its 58 provinces (and 5 municipalities) and in one third of their office branches in the country's 650 districts by mid-2007 [52]. While the organizational unification went well and rapidly, the unification of operations lagged much behind as both spatial and textual data at provincial, district and commune levels remained outdated and thus inconsistent at different levels, and also digitally unlinked [52]. To accelerate the unification of land registration while also developing and modernizing land administration, MONRE outlined a 15-year comprehensive program which included a "Strategy for Information Technology Application and Development for Natural Resources and Environment to Year 2015 with a vision to 2020" approved by the Prime Minister. The Strategy included modernization of land administration by the year 2020 with emphasis on several areas including modernizing the system to collect and update land information and establishing a nationwide unified land database [52]. To implement the program, the government mobilized internal and external financing exceeding US$100 million, including a US$70 million World Bank loan to finance completion of first time land registration, upgrading land rights documentation, completing the unified land registration system and modernizing land registration and administration infrastructure.

As of 2019, Vietnam had developed a unified, comprehensive and decentralized land registration system covering all types of land in the country. At the central level, MONRE has developed and is overseeing reliable procedures and standards for land registration that are being implemented at provincial, district and commune levels. Cadastral data, both textual and spatial, has been updated and is mostly digitally linked with a Land Information System (LIS) software in land registration offices at provincial and district levels that are well-equipped with modern IT equipment. At commune-level, access points with internet connectivity have been established in many of the commune offices.

While a unified land registration system is in place, there are still gaps [58]. It is not comprehensive enough and does not include key areas of land administration such as land valuation and land use planning. Nor does it have links with other relevant computerized information systems within and outside government. To meet these needs, the government is developing a uniform and comprehensive system, the National Land Information System (NLIS), which will include a national land database, a national land information portal and a unified electronic land registration system within a unified framework. It will also include interoperable and standardized data modelling for land data exchange as well as necessary ICT infrastructure to support links with other relevant information systems within and outside government [58].

### 3.2.4. Development Impacts

The implementation of the 1993 Land Law and the issuing of LURCs led to significant increases in agricultural investment [47] and allowed households to pursue non-farming activities [55]. It also increased agricultural productivity and efficiency in the overall economy by promoting the leasing of land to enable its full use as more rural households moved out of farming to more rewarding activities in the evolving non-agricultural sectors of the economy [59].

Over the 34 years of implementation of securing land rights at scale and undertaking other major economic reforms (Moi Doi) and public investments (1986–2019), Vietnam's economy grew at an average annual real growth rate of 6.6 percent as measured by GDP in constant prices, according to calculations based on data from the IMF World Economic Outlook Database [44]. Land registration and tenure security contributed to the growth indirectly mainly by increasing private investment and agricultural productivity while the greater contribution to growth came from the other major economic reforms and public investments [60]. The rapid and sustained economic growth led to reducing the proportion of the population living below the poverty line (US$1.90 per day) from 51.9 percent in 1992

to 1.9 percent in 2018, according to data from the World Bank Poverty and Equity Data Portal [45].

## 4. Discussion

This evaluation of land registration experiences of China and Vietnam has produced notable results and lessons that could help other developing countries to register land at scale. To strengthen the lessons and conclusions from the review while also geographically broadening their global relevance, the findings from the review are supplemented by those from previous evaluations in Africa, notably for Ethiopia and Rwanda, which have had considerable success in registering their land at scale. The discussion below focuses on selected findings to draw lessons especially from the emerging land registration approaches that are cost-reducing, affordable, fast and scalable.

### 4.1. Flexible Legal Frameworks

**Flexible and responsive legal frameworks enabled the adoption of successful approaches to secure land rights at scale.** While initial legislations in both China and Vietnam focused on decollectivizing production, the legal frameworks were flexible enough to allow the adoption of spatial and institutional frameworks as well as new technologies that enabled the registration and certification of land use rights at scale and the efficient operation of land registration systems. In Africa, where Ethiopia and Rwanda have played a lead role in securing land rights at scale, the two countries have also had flexible legal frameworks to accommodate successful registration approaches and efficient operation of land registration systems [8,11,12,61–66].

### 4.2. Land Registration Without a Precise Spatial Framework

**Registration at scale without a precise spatial framework was attained, upgradeable and had positive impacts on investment and productivity.** When they decollectivized their farming and allocated land use rights to households, both China and Vietnam documented land use rights and issued lease certificates to their farmers based mostly on textual data, without a precise spatial framework, in their so-called first round of registration. The certification was done very rapidly to assure farmers of their newly granted land use rights, achieving virtually full national coverage (a large share of 1.5 billion rural arable land parcels in China and about 70 million in Vietnam) within 5–7 years, at a low cost (less than US$1.5 per parcel in the case of Vietnam). Despite lack of a precise spatial framework and the low cost of registration, the certification provided the required tenure security for 20 years (1993–2012) in the case of China, and 15 years (1994–2008) in the case of Vietnam.

Impact evaluation studies found that land certification led to significant increases in tenure security, agricultural investment and productivity in both countries. However, in China, the investment response was undermined by frequent administrative reallocation of land in case of farmers that experienced it. Ethiopia also certified rural land use rights (and most urban land outside informal settlements) without a precise spatial framework over 30 million rural land parcels (during 1998–2009), at a cost of US$1 per parcel [61,65]. Like in China and Vietnam, the certification of rural land in Ethiopia had a significant positive impact on investment and productivity, according to impact studies [62,63]. While documentation of land rights without a spatial framework was a good decision given the urgency to confirm the land rights of millions of households and the difficulties of accessing technology at the time, it was an interim measure which was later followed by the addition of a spatial framework when access to aerial imagery, especially from high resolution satellites and drones, became easier and cheaper. Rwanda took advantage of the accessibility and reduced cost of aerial imagery when it implemented its land registration program as indicated in the next section.

### 4.3. Land Registration with a Precise Spatial Framework

**Registration at scale using mostly orthophotos and high resolution satellite imagery was done cost-effectively**. In their second-round registration, both China and Vietnam upgraded their approach to registration of rural land use rights by adding a spatial framework based mostly on orthophotos (0.4 m resolution) or high resolution satellite imageries (0.5 m resolution), supplemented by ground surveys where physical land boundaries were not visible or land values were high enough to justify the cost. China piloted the second round registration in 2009–2012, scaled it up from 2013 and used it to register 89 percent of all the contracted rural lands by the end of 2018, within five years of scaling up. Vietnam, on the other hand, started implementing second round registration of rural land around 2009, and had, within 10 years, issued land use certificates for around 97 percent of its land as of September 2018.

While information on the average cost of registration per parcel is not available for China and Vietnam, in the case of Ethiopia which has been implementing second level certification of rural land since 2014 and has demarcated 20 million land parcels in 6 years using a similar spatial framework, the average cost per parcel has been around US$8.5 [66]. Rwanda, which registered the entire country's rural and urban land within five years (2009–2013) using only orthophotos (0.4 m resolution) and high-resolution satellite imageries (0.5 m resolution), did so at an average cost of US$8 per parcel [12]. It must be noted that, to register land at scale, rapidly and at a reduced cost, all the four countries maximized the use of orthophotos and high resolution satellite imageries (0.4 or 0.5 m resolution) as well as non-professionals, trained for a short time. While doing so, they minimized the use of high accuracy ground survey instruments, and used land professionals sparingly mainly to provide training and supervision of non-professionals and the management of processes [12,21,31,46,57,66].

Registration of urban land, in both China and Vietnam, was sporadic, and not systematic. It was initially done using low quality cadastral maps, either of inadequate scale, incomplete or outdated. The cadastral mapping was later improved with updated maps of larger scales of 1:500 to 1:1000 in the larger cities and 1:2500 in the smaller cities [34,56]. In Africa, Ethiopia has followed a similar trend while Rwanda registered all its rural and urban land systematically using cadastral maps generated from aerial imageries (0.4 or 0.5 m resolution) at an average cost of US$8 per parcel [12,66].

### 4.4. Unified Land Registration Systems

**Unified registration systems which are efficient, transparent and protective of data quality have been mostly developed**. Fragmented registration systems had not only been costly to build and maintain; they had also been cumbersome and time-consuming. For example, both China and Vietnam had used an approach in which land was registered in one registration system while the attached buildings were registered in a separate system. In addition, China had used separate registration systems for farmland (together with grassland), woodland, forestland and urban land. The development of unified registration systems, involving organizational, operational and digitization components, took about 15 years from vision to full implementation in both Vietnam and China. Much of the implementation was completed by end of 2018 in the case of China while in Vietnam it was still on-going as of November 2020. The design and implementation involved flexibility in legislating, organizing, staffing and in designing land information systems. Implementation required three things: consolidating the responsibilities of real property registration under one organization; integrating or linking real property data bases under one information platform for easy sharing within and outside government; and improving business processes and services. The most challenging activities were the development of national digital land information platforms and systems to support the integration or linking and sharing of real property registration data that, in the case of China, brought together the different registration data for buildings, urban land, farmland, grassland, woodland and forestland of varying quality. But due to experiences gained from earlier

implementation initiatives, development and implementation of a national unified real property registration system integrating or linking all real property registration data was done within the planned time frame (end of 2018) at least for China. Vietnam on the other hand completed implementation of the unified registration system but is still developing a broader land administration information system to accommodate all core land administration processes as well as links with other information systems within and outside government.

As for the two African countries, Rwanda possesses a well-developed digital land administration information system with links to other information systems within and outside government. On the other hand, Ethiopia has developed one for rural land but has ways to go in developing another for urban land and links between them, or integrating the two to form a unified national land administration information system [12,64].

### 4.5. Development Contributions of Secured Land Rights

**Reforms and investments to secure land rights at scale contributed to high economic growth and a rapid reduction of poverty**. In all the case study countries, previous evaluation studies confirmed positive impacts of the land tenure reforms (legal clarification and certification of rights) on land tenure security, investment and productivity except for Rwanda where the productivity response time was too short to allow quantifiable results [12]. But context matters. The land tenure reforms in all the case study countries were part of broader economic reforms to transform from central planned to market-based systems (in China, Vietnam and Ethiopia) or, in the case of Rwanda, to resettle, rebuild and recover from civil war, displacement and genocide [12]. The countries grew their economies strongly over the periods of land tenure reform, with China, Vietnam, Ethiopia and Rwanda recording average annual real economic growth rates of 9.4%, 6.6%, 8.2% and 7.7%, respectively, calculated using data from the IMF World Economic Outlook Database [44]. As earlier noted, while land tenure reforms contributed indirectly to the growth particularly through enhanced investment incentives and productivity, a much greater contribution came from complementary economic reforms and public investments as documented for China by Garnaut, Song and Fang [43], for Vietnam by Le [60], for Ethiopia by World Bank [67] and for Rwanda by Crisafulli and Redmond [68].

The combined impact of land tenure and other major economic reforms and public investments on poverty reduction was equally impressive, with China and Vietnam mostly eradicating extreme poverty while Ethiopia and Rwanda reduced considerably the proportion of people living below the poverty line (US$1.90 per day) from 72.3 percent in 1995 to 32.6 percent in 2016 and from 69 percent in 2005 to 56.5 percent in 2016 respectively, according to data from the World Bank Poverty and Equity Database [45]. Land rights for women were enhanced at least in the cases of Ethiopia, Rwanda and Vietnam. Ninety (90) percent of land rights certificates were issued to women in sole or joint ownership in Ethiopia [66], 68 percent in Rwanda [12] and 62 percent in Vietnam [57].

### 4.6. Lessons Learnt and Challenges

**There are important lessons to learn from these Asian and African land registration experiences.** At least five lessons stand out from the land registration experience of China and Vietnam, reinforced by the more recent experience of Ethiopia and Rwanda, to secure land rights at scale.

1. **Strong political commitment**. Political commitment is crucial to securing land rights at scale. As land registration was part of fundamental economic reforms to move from a socialist planned economic system to a market-based economic system in China, Vietnam and Ethiopia and part of comprehensive reforms to recover from civil war and genocide in Rwanda, there was strong political commitment to land registration from the highest to the lowest level of government. The political commitment carried the day.

2.  **Flexible legal framework**. Flexibility in legal frameworks facilitates the adoption of some emerging registration approaches that are fast, affordable and scalable to register land rights for all.

3.  **Registration based on imprecise spatial framework**. Registering land rights mostly on the basis of textual data, without a precise spatial framework, and with limited use of land professionals is a low-cost, affordable, scalable and upgradable approach best suited for first round registration especially for low value rural land as was demonstrated by China, Vietnam and Ethiopia in the 1990s and 2000s. But with rapidly declining costs and easier access to aerial imagery, the option of adding a spatial framework (based on aerial imagery) to textual data is increasingly becoming realistic.

4.  **Registration using mostly aerial imagery for a spatial framework**. Adding a spatial framework based mostly on orthophotos and high resolution satellite imagery as an upgrade from first round registration has proven to be a scalable cost-effective land registration approach as it minimizes the use of expensive ground survey equipment while maximizing the use of non-professionals (with short training) in place of surveyors and land lawyers to demarcate and adjudicate land. In fact, Rwanda covered the registration of all its rural and urban lands using only aerial imagery (orthophotos of 0.4 m resolution and satellite imagery of 0.5 m resolution) while China, Vietnam and Ethiopia supplemented aerial imagery with ground surveying only in areas where either physical boundaries were not visible or land values were high with a potential to cause contestation of land boundaries.

5.  **Developing a unified digital registration system**. Developing a unified digital registration system is feasible and improves efficiency, transparency, protection of data quality and integrity, and facilitation of information sharing. Rwanda has done it and it has been beneficial. China, Vietnam and, to a less extent, Ethiopia (for rural land) have also done and it has started to pay off.

There are also at least three challenges.

1.  **Keeping land registration data updated.** In Vietnam, Ethiopia and Rwanda, registration of subsequent transactions after the first round of certification was difficult because registration forms used in the latter were in a format that could not allow the recording of subsequent transactions. In the second round of land certification (the tail end of first registration in the case of Rwanda), the forms were redesigned and digitized, and systems were developed to enable registration of subsequent transactions. Another challenge related to the maintenance of land registration, at least in the case of Rwanda where the issue was closely monitored, is that registration of subsequent land transactions has remained relatively low for rural land mainly because registration charges have been high relative to the value of land. The government of Rwanda has been considering options to address it including reducing registration charges for rural land while increasing those for urban land to effect cross-subsidization since urban land is of higher value. In China and Vietnam, there are no reported issues of registration of subsequent transactions presumably because land values in both rural and urban areas are high enough to cover land registration charges. As for Ethiopia whose land registration coverage includes a lot of rural low-value land, the registration of sub-sequent land transactions has not been an issue so far presumably because many of the rural land offices have not started charging registration fees given that the new national rural land administration information system is still being installed and many land offices have not been covered yet [64].

2.  **Addressing land tenure insecurity and informality in urban slums**. While registration of urban land in the four case study countries has gone well, notwithstanding the poor quality of cadastral mapping in the case of Ethiopia [65], the formalization of informal settlements and registration of land rights in urban slums are still inadequate despite great improvements made by Vietnam and China. For example, the percentage of the urban population living in slums declined from about 44 in 1990

to about 25 in 2018 in China and from 61 in 1990 to 14 in 2018 in Vietnam compared to the decline from 96 in 1990 to 64 in 2018 in Ethiopia and from 96 in 1990 to 42 in 2018 in Rwanda [36]. The prevalence of slums especially in the African countries of Ethiopia and Rwanda remains a serious development challenge.

3.  **Registering customary land rights**. While China and Rwanda have virtually no customary or communal land tenure systems left, Vietnam and Ethiopia still have them but the latter has successfully piloted approaches which are being scaled up to register land rights especially for pastoralist groups [69]. Vietnam, on the other hand has registered most of the customary lands which are found in the mountainous and forest areas occupied by indigenous ethnic communities which account for about 13 percent of Vietnam's total population [70]. Vietnam's customary land tenure systems are recognized under state laws and, before the lands are registered and land use rights certificates (LURCs) issued, the land-owning communities are organized into legal entities on the basis of Bylaws [70]. The main challenge customary land tenure systems face in Vietnam, like in many other countries especially in Africa, is how to organize the land-owning groups into legal entities and to strengthen their capacity to manage their land, forestry, pastoral and other natural resources [5].

## 5. Conclusions

When China and Vietnam decollectivized agricultural production and allocated rural land to farming households in the 1980s and 1990s, they were faced with a formidable task of registering land rights covering about 1.5 billion rural arable land parcels in China and about 70 million in Vietnam. This was in addition to the need to registering urban land rights. In both countries, the registration of rural land was done in two rounds. The first round was done rapidly and much of it within 7–8 years after land allocation, covering about 90 percent of the allocated land parcels in both countries. While the registration was done at county level in China and at district level in Vietnam, the field work of adjudication and demarcation was done mainly by commune authorities (and peoples committees in Vietnam) after short training done by land professionals from land administration departments, who also managed the registration processes. The processes were participatory involving not only the holders of land use rights but also the neighbors. They were also cost-effective, costing in the case of Vietnam, about US$1.5 per parcel.

The second round in China started with piloting from 2009 to 2012, and scaling up from 2013 to 2018 while in Vietnam, implementation of the second round started in 2009 and was completed around 2018 as well. The second round was done in both countries mainly to add or upgrade cadastral maps and to digitize and develop unified land registration systems. The cadastral maps were based mostly on orthophotos and high resolution satellite imageries. While the field work of adjudication and demarcation was done by local non-professionals mainly commune leaders and peoples committees supported by land professionals (called cadastral officers in Vietnam) from land agencies, the documentation and digitization was done only by government staff from land agencies supplemented by contracted staff, in the case of Vietnam. The work in Vietnam involved 12,000 cadastral staff from government and many more in the case of China. When the second round of registration ended in 2018, China had registered 89 percent of the contracted rural lands while Vietnam had issued land use rights certificates covering about 97 percent of its rural land. Grassland, woodland, forestland, collectively owned land and state lands were all registered in both countries. In addition, urban land was registered over time and completed by 2018 in both countries. The registration of urban land was based on applications from holders of land use rights (sporadic method), and the cadastral maps used were updated over time.

The land registration processes in both China and Vietnam were underpinned by flexible legal and spatial frameworks that accommodated the technologies and standards used during the two rounds of registration. They were also participatory involving the holders of land rights, their neighbors and local leaders. In addition, they were inclusive,

enabling the registration of rights of all landholders. The registration processes were also upgradable and scalable, and used skilled land professionals and costly survey equipment sparingly while maximizing the use of non-professionals and aerial imageries. In short, the registration approaches used were flexible, participatory, cost-reducing, affordable, fast, inclusive and scalable. These are the same kind of principles that underpin the fit-for-purpose land administration approach, notwithstanding the fact that they were alluded to in the VGGTs in 2012 and used long before the approach was formalized in the guiding principles for country implementation published in 2016 [9]. Rwanda, which completed its land registration program recently, and Ethiopia, whose land registration program is ongoing, have applied similar principles with success, as indicated in the last chapter. In all the four countries, there was strong political commitment from government from central level down to local level, and land registration was part of comprehensive economic reform programs. The findings from this review of land registration experience in China and Vietnam, buttressed by those from previous reviews for Ethiopia and Rwanda, suggest that developing countries can secure their land rights at scale within a generation if they adopt similar registration approaches. They also suggest that the fit-for-purpose land administration guiding principles for country implementation indicated above [9] would be useful for developing countries considering to engage in providing secure land rights at scale.

It should be noted that in China, Vietnam and the two African countries of Ethiopia and Rwanda, previous evaluation studies confirmed positive impacts of the land registration programs on investment and productivity except for Rwanda where the productivity response time was too short to allow quantifiable results. While the relationship between land tenure security and economic growth is indirect and its measurement has been confounded by attribution problems [19], the positive contributions of land tenure to private investment and agricultural productivity suggest that the land registration programs together with the other major economic reforms and public investment in the four countries contributed considerably to the economic development of these countries. During the periods when land registration programs were implemented, the economies of China, Vietnam, Ethiopia and Rwanda recorded average annual real economic growth rates of 9.4%, 6.6%, 8.2% and 7.7%, respectively. The impact on poverty reduction was equally impressive, with China and Vietnam mostly eradicating extreme poverty while Ethiopia and Rwanda reduced considerably the proportion of people living below the poverty line (US$1.90 per day) from 72.3 percent in 1995 to 32.6 percent in 2016 in the case of Ethiopia, and from 69 percent in 2005 to 56.5 percent in 2016 in the case of Rwanda. In addition, land rights for women were enhanced at least in the cases of Ethiopia, Rwanda and Vietnam. Ninety (90) percent of land rights certificates were issued to women in sole or joint ownership in Ethiopia, 68 percent in Rwanda and 62 percent in Vietnam.

The experience of these Asian and African countries in securing their land rights at scale offers lessons that other developing countries can apply to secure land rights for all in one generation. The most two crucial lessons are: political commitment to securing land rights at scale; and using flexible legal and spatial frameworks that fit the purpose of land registration, instead of the rigid technical standards set by land professionals. However, there are also three challenges to note from the experience of these four countries. The first challenge is keeping land registration records updated. This is an important issue in Rwanda where registration of subsequent land transactions has remained relatively low for rural land mainly because registration charges have been high relative to the value of land. The government of Rwanda has been considering options to address it including reducing registration charges for rural land while increasing those for urban land to effect cross-subsidization since urban land is of higher value. Keeping land records updated is critical to the sustainability and integrity of land registration systems.

The second challenge is addressing urban slums, notwithstanding the success achieved in registering urban land. The percentage of the urban population living in slums in 2018 was 25 for China, 14 for Vietnam, 64 for Ethiopia and 42 for Rwanda. The prevalence

of slums especially in the African countries of Ethiopia and Rwanda remains a serious development challenge. The third challenge is registering customary and community land rights. Among the four countries covered by this review, China and Rwanda have virtually no customary land tenure systems remaining while Vietnam and Ethiopia still have them, and they are recognized under state laws. In Vietnam, customary communities mostly in the mountain areas have been formalized and issued land use rights certificates covering their agricultural and forest lands. The challenge they face, like many other customary tenure communities especially in Africa, is how to organize themselves into legal entities and to strengthen their capacity in managing their land, pastoral, forestry and other natural resources.

The persistence of the three challenges highlighted above, namely, keeping land registration data updated, addressing land tenure insecurity and informality in urban slums and registering customary land rights, suggests a need for intensifying evaluation of implementation experiences in these areas to learn lessons that can help in addressing these challenges more effectively.

**Funding:** This work received no external funding.

**Institutional Review Board Statement:** Not applicable.

**Informed Consent Statement:** Not applicable.

**Conflicts of Interest:** The author declares no conflict of interest.

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
