# Peer review of "Experiences and Development Impacts of Securing Land Rights at Scale in Developing Countries: Case Studies of China and Vietnam"

_land, doi:10.3390/land10020176_

Round 1
Reviewer 1 Report
I can´t cut and paste into this box. Instead I have enclosed a file with the review comments, see below this box.
What do you want to do ? New mailCopy
What do you want to do ? New mailCopy What do you want to do ? New mailCopy

Author Response
Response to General Comments
In section 1, the suggestion is to include in section 1 that the principles used in the registration processes in China and Viet Nam are similar to those underpinning the FFPLA and yet were applied by those countries before the FFPLA Guidelines were developed. This has been done.
A related suggestion is to include in the Conclusion Section that “the success of the outcome in China and Viet Nam could then be seen as underpinning the FFPLA Guidelines as useful for developing countries considering to engage in providing secure land rights at scale”. This has been included in Conclusion Section as suggested.
In Section 2, the suggestion is to provide “a more clear identification of the research problem and the methodology applied to address the problem through scientific analysis”, and the choice of the 4 areas of assessment explained. I have addressed this by explicitly stating 3 research problems the research would address and tied this to the 4 areas of assessment. I have also stated the research methodology, basically, qualitative desk analysis based on case studies, and explained specifically how the assessment of impacts on land registration on development impacts would be done, in a two-stage process, first relating land registration to investment and productivity (a direct relationship) and second relating land registration to economic growth (an indirect relationship) going through investment and productivity as intermediate outcomes.
In section 3, it is suggested to provide illustrations of spatial frameworks and organizational structures for China and Viet Nam. Due to lack of clean cadastral maps and organograms, I have provided instead map scales for cadastral maps and resolutions for orthophotos and HRS imageries (and references). For organizational structures, I have provided references of accessible documents where organograms can be found.
In addition, number of land parcels were requested. These have been provided, notably 1.5 billion land parcels for China, and 70 million for Viet Nam, although it was difficult getting specific numbers for break-downs of registered numbers of parcels especially for China. Where numbers of parcels are unavailable, percentages of registered parcels have been provided.
In Section 4, it is a compliment that 4 areas of assessment in chapter 3 as useful. This is gracefully noted and appreciated.
In Section 5, suggestion to include in the Conclusion chapter more detailed discussion of the economic benefits of developing the registration systems, noting that the benefits may also be related to other policies and public investments (as a caveat). This has been done and relevant
references provided, especially highlighting the indirect relationship of land registration to economic growth and the contributions of other economic reforms and public investments to the benefits of land registration. It is not only done in the concluding chapter but more so in the Discussion Chapter.
The suggestion to note that the case study countries (except Rwanda which is unique in a way as it was recovering from genocide and civil war) are all centrally planned economies has been done, not under Conclusion Chapter alone (highlighting importance of political commitment due to centrality of reforms from socialist economic systems) but also under Assessment and Discussion chapters highlighting relevance of socialist political systems to the participatory processes used in land registration.
It was suggested to address issue of maintenance (updating and upgrading) which has been a problem in Rwanda; this has been done not only in concluding Chapter but also Discussion Chapter (I also highlighted the challenges of addressing slums in all 4 countries as well as customary/community land rights especially in Viet Nam (and Ethiopia) in the Discussion and Concluding chapters.
Finally, it was suggested to refer more specifically to the research problem and show that the review indicates the that recent FFPLA Guidelines are well founded in the experiences and development impacts from the processes undertaken in China and Viet Nam for providing secure land rights at scale. This has been done in the Conclusion Chapter and tied to Section 2 on the research problem.
Line 2-4 to include specifically China and Viet Nam in the title – this has been done, and the title changed.
Line 22 – include key words to be better aligned with those listed in the website – done by adding fit-for-purpose land administration.
Line 53 – reference 9 is on FFPLA Country Implementation, not pilots. Changed accordingly.
Line 85 – references missing. They have been added, and table modified, in light of change in title to focus on China and Viet Nam, and to include estimates of total land parcels in each of the 4 countries.
Line 155- Meaning of competitive mechanisms – replaced by auctions.
Line 167 – Certificate of title not correct – changed to land use rights certificates, also in lines 175 and 196.
Line 218- indicate size in ha or square km. Done in ha.
Line 299-304, economic growth not just due to land registration alone – have indicated other factors as well and also highlighted that impacts of land registration are indirect and unquantifiable; this has been reflected through the article also in chapters 4 and 5.
Line 304 – references missing on World Bank poverty Data Portal. Has been provided.
Section 4.2. May be also mention about situation in Rwanda. Done
Section 553-557. Indicate extreme poverty reduction more specifically. Done in numbers both in the Discussion and Conclusion chapters.
Thank you for your valuable comments.

Reviewer 2 Report
General comments:
This is a valuable paper, and the contribution of the land tenure case studies of China and Vietnam are significant as the stories have not been told in such detail before.
The FFP methodology has only recently been formalised and guidelines produced. However, it is clear from the Chinese case study, especially, that key principles of the FFP methodology were adopted many years ago. This should be emphasised in the paper.
The methodology description needs to be strengthened with a clearer statement of the research problem and how it will be responded to.
Illustrations would improve the paper and bring it alive for the reader. Examples of maps, sketches and quality improvements over time could be included, for example.
These have been some of the largest land tenure programs in the world. More statistics, for example, on the number of parcels and resources involved in various stages of implementation should be included to emphasise the scale of the problem being solved.
The political systems of the four countries should be described and how the political systems may have influenced the land tenure programs and their accelerated implementation. Top level political support is a key success factor.
Solutions need to be sustainable and effective maintenance regimes need to be established. This should be highlighted in conclusions. Rwanda is an example of where this went wrong.
Where possible, the specific contribution of security of tenure to economic growth and poverty reduction should be stated. Yes, security of tenure has been a contributary factor, but by how much? Need some quantitative statistics.
Many developing countries have to solve the problems of informal settlements and customary tenure practices in their land tenure programs. If these have not been a major problem in the four countries reviewed then this should be stated. Otherwise, the solution to these issues should be included.
Some sentences are very long and complex and are difficult to interpret. They should be broken into shorter sentences and commas used correctly, e.g. before 'which'.
Detailed comments:
Some of the terminology needs to be checked and corrected:
- 'Leasehold contract' rather than 'title'
- 'Legal and regulatory framework' rather than 'legal framework'
- 'Land registration and cadastre' rather than 'land registration'
A list of recommended key words were provided to authors to create consistency across the FFPLA papers. Check to make sure they are used , where appropriate.
Line 48: state land is also a major challenge.
Line 53: this GLTN/UN-HABITAT reference [9] is not a pilot, but a set of guidelines for country implementation.
Line 84: Table needs to be reformatted to comply with template.
Line 139 -140: ‘In 2013, the central authorities advanced the target of registering rural land rights to five years.’ Needs clarification.
Line 235: what is ‘appropriate remote sensing’? Provide examples of what was used.
Line 276 - 282: A number of different terms are used: ‘a national computerized land information platform’ / ‘unified real estate database’ / ‘national unified property registration database’. Are these different or the same solution?
Line 347: illustrations included of ‘sketch and cadastral maps’ would be useful.
Line 379: ‘spatial data based mostly on orthophotos and high resolution satellite imagery’ What scales and resolutions were used?
Line 390 – 393: Who carried out this work? Were they qualified surveyors or trained citizens and how many?
Line 465: is ‘contracting’ correct?
Line 476: ‘US$1.5’ should be ‘US$1.5/parcel’?
Line 470 – 486: ‘Registration at scale without a spatial framework was attained, upgradeable and had positive impacts on investment and productivity’. At the time, this was a good decision. However, with current ease of access to aerial imagery through drones and satellites, for example, this is not a strategy that should be recommended.
Line 509 and Line 511: explain what a ‘cadastral index map’ is or provide a reference.
Line 526: ‘integrating land data under one information platform’ One information platform can have an ICT architecture of many interoperable information platforms that looks like one information platform to the user.
Line 527: day to day
Line 542 – 557: It would be good to know if there were complementary programs to the land tenure programs in these countries that focused on financial services and provision of mortgages, development of land markets, capacity development… that accelerated the benefits?
Line 563 – 567: The second lesson learnt is not a good recommendation in today’s context of ease of access to aerial imagery (see comment above).
Author Response
Response to General Comments
This is a valuable paper – thanks.
Suggestion to emphasize that the FFP methodology has only recently been formalized and guidelines produced and yet the review of China and Viet Nam indicates that similar principles were applied retrospectively. This has been done in Introduction, Discussion and Conclusion chapters.
Needs to strengthen methodology description with clearer statement of the research problem and how it will be responded to. Done by specifying 3 research problems and methodology of addressing them through qualitative desk research with case studies. This section has been overhauled including describing a methodology for estimating development impacts.
Improve paper with illustrations for maps etc. to make it clearer to the reader. Due to lack of clean cadastral maps and organograms, I have provided instead map scales for cadastral maps and resolutions for orthophotos and HRS imageries (and references for the reader). For organizational structures, I have provided references of accessible documents where organograms can be found.
Since are largest land tenure programs, include statistics, eg. on land parcels. These have been provided, notably 1.5 billion land parcels for China, and 70 million for Viet Nam, although it was difficult getting specific numbers for break-downs of registered numbers of parcels especially for China. Where numbers of parcels are unavailable, percentages of registered parcels have been provided.
Describe political systems and how these could have influenced implementation of land tenure programs. This has been done including indication that the case study countries (except Rwanda which is unique in a way as it was recovering from genocide and civil war) are all centrally planned economies reforming toward market-based economies. It has been done in Chapters 3 to 5 not only under Conclusion Chapter alone (highlighting importance of political commitment due to centrality of reforms from socialist economic systems) but also under Assessment and Discussion chapters highlighting relevance of socialist political systems to the participatory processes used in land registration.
Suggestion to highlight effective maintenance and sustainability especially in conclusion, reference Rwanda. This has been done not only in Conclusion chapter but more so in the Discussion Chapter.
Specify contributions of land tenure security to economic growth and poverty reduction including quantitative statistics. I have dealt with this at length including in Methodology, Discussion and Conclusion Chapters by indicating that: (i) the relationship between land tenure security and economic growth is an indirect one, through investment and productivity (as intermediate outcomes) that has been estimated; hence the contributions to economic growth cannot be quantified (past attempts to do so have failed mainly because of attribution problems. It can only be qualitatively described and its strength related mainly to the impacts on investment and productivity. Relevant references have been provided.
Discussion issues of informal settlements and customary tenure practices. The former is a problem in all the 4 countries and the latter is a problem mainly in Ethiopia and Viet Nam. These have been discussed (together with the problem of maintenance of land registration systems) as challenges both in the Discussion and Conclusion chapters.
Some sentences are long and complex and difficult to interpret. I have revised them to break them down into shorter sentences and improved the use of words and commas as suggested.
Detailed Comments
Suggestion to check terminologies:
ď‚· leasehold contract rather than title – have dropped the use of “title”, and used contractual land use rights or land use rights.
ď‚· legal and regulatory framework – have used legal framework
ď‚· land registration and cadaster – have used land registration.
Line 48 – state land is also a major challenge. Included it.
Line 53 – Reference [9] not pilot but guidelines for country implementation – corrected.
Line 84 – Table needs reformatting according to template – table changed and references added.
Line 139 -140. Advancing the target of registering land to 5 years – corrected. It was a new target set for completion of land registration program.
Line 235 – what is appropriate remote sensing – I specified it with aerial imagery and relevant resolutions.
Line 276-282. Number of conflicting terms used – corrected and limited them to: digital land information platform; and unified real property registration system.
Line 347 – illustrations including sketch and cadastral maps would be useful – since I could not get clean illustrations, I provided cadastral map scales and accessible references for the reader.
Line 379 – spatial data – orthophotos and HRS imageries – resolutions of 0.4 meters and 0.5 meters respectively, have been included.
Line 390- 393. Who carried out the work – Qualified surveyors (called cadastral staff) from the land agency and contracted staff trained and supervised members of the peoples committees and communes to do field demarcation and adjudication. Document processing and digitization was done by professional land agency staff.
Line 465 – contracting not correct – it is registration.
Line 476 – it it is US$1.5 per parcel as suggested.
Line 470-486 – Comment that use of textual data without a spatial framework may have been a good decision then but has been overtaken by easier access to cheaper aerial imagery – correct. Have adjusted accordingly.
Line 509 and 511 – cadastral index map – it is a special kind of a cadastral map – have replaced it with cadastral map derived from aerial imagery to avoid confusing the reader.
Line 526 – One information platform and ICT architecture of many interoperable information platforms – I have changed the terminologies to clarify that the national information platform is for purposes linking different systems and databases and the sharing of data.
Line 527 – day to day. I deleted it.
Line 542-557 – Were there complementary programs to the land registration program – Yes, it was part of comprehensive economic reforms and public investments in the 4 countries. I have clarified this in Methodology, Discussion and Conclusion chapters.
Line 563-567 – Second Lesson learnt is not a good lesson in todays easier access and cheaper aerial imagery – correct. I have modified the lesson.
Thank you for your valuable comments.

Round 2
Reviewer 1 Report
What do you want to do ? New mailCopy What do you want to do ? New mailCopy What do you want to do ? New mailCopy What do you want to do ? New mailCopy

Author Response
The suggestion to start with a heading, in bold, for each lesson and challenge in Section 4.6, has been done exactly as suggested
Reviewer 2 Report
The revisions are excellent and strengthen the paper considerably. Just one edit required:
Line 797: 'These are the same principles that underpin the fit-for-purpose land administration approach, notwithstanding the fact that they were used long before the approach was formalized in the VGGTs in 2012 and its guiding principles for country implementation published in 2016.'
The FFP approach was not explicitly mentioned in the VGGTs as such, since the term was used for the first time in the FIG/WB publication on FFPLA in 2014 and the approach expanded in 2016. So maybe the sentence line should read:
'These are the same kind of principles that underpin the fit-for-purpose land administration approach, notwithstanding the fact that they were alluded to in the VGGTs in 2012 and used long before the approach was formalized in the guiding principles for country implementation published in 2016 (9).'
Author Response
Reviewer 2 suggested an edit to the sentence on line 797 along the following lines:
"These are the same kind of principles that underpin the fit-for-purpose land administration approach notwithstanding the fact that they were alluded to in the VGGTs in 2012 and used long before the approach was formalized in the guiding principles for country implementation published in 2016 [9]".
The suggestion was adopted word by word.